# Anti-Inflammatory Potential of Pteropodine in Rodents

**DOI:** 10.3390/metabo13080907

**Published:** 2023-08-03

**Authors:** Rogelio Paniagua-Pérez, Laura Sánchez-Chapul, Eduardo Madrigal-Bujaidar, Isela Álvarez-González, Eduardo Madrigal-Santillán, Lidia Cruz-Hernández, Carlos Martínez-Canseco, Celia Reyes-Legorreta, Lidia Ruiz-Rosano, Cecilia Hernández-Flores, Rene Valdez-Mijares, Alejandra Quintana-Armenta

**Affiliations:** 1Laboratorio de Bioquímica, Instituto Nacional de Rehabilitación “Luis Guillermo Ibarra Ibarra”, Mexico City 14389, Mexico; lcruz@inr.gob.mx (L.C.-H.); cmartinez@inr.gob.mx (C.M.-C.); liruiz@inr.gob.mx (L.R.-R.); chernandez@inr.gob.mx (C.H.-F.); rvaldez@inr.gob.mx (R.V.-M.); aquintana@inr.gob.mx (A.Q.-A.); 2Laboratorio de Enfermedades Neuromusculares, División de Neurociencias Clínicas, Instituto Nacional de Rehabilitación “Luis Guillermo Ibarra Ibarra”, Mexico City 14389, Mexico; 3Laboratorio de Genética, Escuela Nacional de Ciencias Biológica-Instituto Politécnico Nacional, Mexico City 11340, Mexico; emadrigalb@ipn.mx (E.M.-B.); ralvarez@ipn.mx (I.Á.-G.); 4Laboratorio de Medicina de la Conservación, Instituto Politécnico Nacional, Escuela Superior de Medicina, Mexico City 11340, Mexico; emadrigals@ipn.mx; 5Laboratorio de Neuroprotección, Instituto Nacional de Rehabilitación “Luis Guillermo Ibarra Ibarra”, Mexico City 14389, Mexico; cereyes@inr.gob.mx

**Keywords:** anti-inflammatory assays, murine model, rodent-induced edema

## Abstract

Pteropodine (PT) is a component of some plants with potentially useful pharmacological activities for humans. This compound has biomedical properties related to the modulation of the immune system, nervous system, and inflammatory processes. This study addresses the anti-inflammatory and antioxidant capacity of pteropodin in a murine model of arthritis and induced edema of the mouse ear. To evaluate the anti-inflammatory activity, we used the reversed passive Arthus reaction (RPAR), which includes the rat paw edema test, the rat pleurisy test, and a mouse ear edema model. The antioxidant effect of PT was evaluated by determining the myeloperoxidase enzyme activity. PT showed an anti-inflammatory effect in the different specific and non-specific tests. We found a 51, 66 and 70% inhibitory effect of 10, 20 and 40 mg/kg of PT, respectively, in the rat paw edema test. In the pleurisy assay, 40 mg/kg of PT induced a low neutrophil count (up to 36%) when compared to the negative control group, and 20 mg/kg of PT increased the content of lymphocytes by up to 28% and the pleural exudate volume decreased by 52% when compared to the negative control group, respectively. We also found an 81.4% inflammatory inhibition of the edema ear with 0.04 mg/ear of PT, and a significant myeloperoxidase enzyme inhibition by the three doses of PT tested. We conclude that PT exerted a potent anti-inflammatory effect in the acute inflammation model in rodents.

## 1. Introduction

Some diseases, such as rheumatoid arthritis (RA), are characterized by chronic inflammation that affects mainly the joints, where it can produce a progressive degree of deformity and functional disability related to the continuing destruction of cartilage, as well as damage to tendons, ligaments, and bones [1]. In addition, the disease can affect other organs, such as the eyes, lungs, pleura, heart, skin, and blood vessels [2]. In general, the average prevalence is 1%, although some specific studies have shown higher levels, probably related to race, diagnostic criteria, and methodological differences [3]. The reported incidence is variable; for example, a high annual incidence rate of 90 cases/100,000 habitants was reported in Germany [4,5], whereas values of 42 and 45 cases/100,000 were found in Finland and Japan [6,7], respectively. In Mexico, about one million people have some degree of illness [8].

Although the precise etiology of RA is not entirely known, the leading role played by self-immunity in its development is well documented, as well as its genetic predisposition, alterations in the union of joints or cartilage, joint injury, certain bacteria, fungi, or viruses that infect the joints, among others [9]. Painful inflammation is normally a restorative response that sometimes progresses towards a chronic situation that usually gives rise to degenerative arthritis. In the inflammatory process of RA, there are cellular and humoral components involved in the etiology of the disease; however, the increase in the pro-inflammatory cytokines such as IL-8, TNF-alpha, IL-6, IL-1beta, IL-15, IL-18, and IL-17A, which can be detected in both synovial fluid and serum of patients, play the main role [9].

Some of the drugs used to reduce disease progression and improve the quality of life are chloroquine, hydroxychloroquine, sulfasalazine, D-penicillamine, azathioprine, cyclophosphamide, methotrexate, cyclosporine, leflunomide, and some steroid and non-steroidal drugs [10,11,12]. However, the therapeutic effect is usually symptomatic, nonpermanent, and can cause collateral damage. The group of immunosuppressants can cause numerous adverse reactions; however, their mechanisms are not yet well known. Some collateral damages of steroids are osteoporosis, predisposition to infections, gastro toxicity, increased risk of skin infections, such as bacterial (e.g., cellulitis) and fungal (e.g., tinea, candidiasis), skin thinning, resulting in easy bruises (purples), skin tearing after minor injury, and slow healing; these effects are most prominent on sun-exposed areas, particularly the back of the hands and the forearms [13,14]. Other steroids-induced alterations are stretch marks (striae), particularly under the arms and in the groin, acne, clusters of small spots on the face, chest, and upper back, excessive hair (hypertrichosis), hair loss (alopecia), and subcutaneous lipoatrophy (loss of fat under the skin surface) caused by an injected steroid that does penetrate deep enough into the muscle [15,16,17]. This makes the search for new compounds that efficiently reduce the physiopathological mechanisms that lead to the disease and the toxic side effects worthwhile. To achieve these purposes, extracts or compounds derived from plants have been investigated using the reversed passive Arthus reaction (RPAR) in rabbits to detect more effective anti-inflammatory agents for the treatment of RA [18,19]. Animal murine models used in the investigation of inflammatory processes are as diverse as the materials and repair strategies. In general, and as in other areas of science, animal studies begin with models such as mice or rats, later moving on to rabbits, pigs, and sheep. Due to the type of tissue, most studies focus on medium- and large-sized animals, since the work area is larger, and this facilitates the surgery to be performed. In addition, the results obtained in these cases are more easily extrapolated to possible clinical use.

*Uncaria tomentosa* (UT) is a Rubiaceae plant native to Peru, commonly called “cat’s claw”, used in traditional medicine to treat some diseases like cancer, arthritis, candidiasis, menstrual and intestinal disorders, and HIV infections [20]. Other investigations have confirmed its biomedical properties with immunostimulant, cytostatic, anti-inflammatory, antimutagenic, and anticancer effects [21,22]. Several components of the plant have been chemically identified as oxindole alkaloids, proanthocyanidins, polyphenols, triterpenes, and sterols, among others [23]. Six oxindole alkaloids have been isolated from the plant, including pteropodine (PT), also called uncarine, which is a heterohimbine-type oxindole that has been reported to show an apoptotic effect in leukemic lymphoblasts and participates in the improvement of memory impairment induced by dysfunction of cholinergic systems in the brains of mice [24,25]. These data suggest that PT could act synergistically with other UT components in one or more of the effects reported for the plant. Quality control of medicinal plants is a multi-step process that covers all stages of production, from the plant as raw material to its packaging as a finished product, whether it is a medicine or herbal remedy. From the point of view of its application, quality is aimed primarily at authenticating the plant species. Knowing the metabolic content of the plants is also important and fundamental because it allows for establishing, which will be the marker that will be used for the pharmacopeial trials. It is strictly necessary that the medicinal plants that are intended for preparing herbal products comply with the specific analytical determinations related to their quality. In Mexico, these determinations are compiled in the Herbal Pharmacopoeia of the United Mexican States (FHEUM), which is the official document used for this purpose. Any medicinal plant that must be used for therapeutic purposes to be marketed, or that affects aspects of the health of the general population, will have to demonstrate its quality in accordance with the official procedures framed in the FHEUM (identity, composition, and purity parameters), as well as in all current regulations related to the regulation of herbal products [20,21,22,23].

Our laboratory evaluated the genotoxic and antigenotoxic potential of PT, finding that this compound is not genotoxic in mice, since PT significantly decreases the frequency of sister-chromatid exchanges and micronucleated polychromatic erythrocytes in mice [26]. Moreover, we determined that PT protects mouse cells from DNA damage induced by doxorubicin (DX), which is an antineoplastic agent that damages DNA; this protective action of PT is due to its efficient free-radical-trapping ability showed in the DPPH assay [27].

Based on the above information, in this report, we expanded the studies on the capacity of PT as an anti-inflammatory agent by applying tests in mice and rat, attempting to determine its anti-inflammatory potential in a mouse model that has the physiopathological mechanisms of rheumatoid arthritis and in a mouse ear edema model.

## 2. Materials and Methods

### 2.1. Reagents and Animals

PT (99% pure) was obtained from Acceso-Lab Chemicals (Mexico City, Mexico). Ibuprofen (IBU), prednisone (PRED), antiserum of egg albumin, Freund’s complete adjuvant, 12-O-tetradecanoylphorbol-13-acetate (TPA), indomethacin (IND), acetone, and ethyl ether were purchased from Sigma Chemicals (St. Louis, MO, USA). Ovalbumin antiserum (rabbit polyclonal to ovalbumin) was obtained from Abcam (Mexico City, Mexico), and NaCl solution (9%) was obtained from Baker S.A. (Mexico City, Mexico). For the assay, we used 8-week-old male Wistar rats weighing 250 g on average and NIH mice averaging 25 g in weight. The animals were obtained from the National School of Biological Sciences (Instituto Politecnico Nacional, IPN, Mexico City, Mexico) and maintained in metal cages at 23 °C in a 12 h light–dark cycle and allowed to consume water and food freely (Lab chow 5001, Purina, Mexico City, Mexico). This protocol was approved by the Ethics and Biosecurity Committee of the National Rehabilitation Institute, SSA, Mexico City, Mexico.

### 2.2. Description of the Murine Model

The Arthus reaction allows us to experimentally reproduce the local physiopathology of RA. The obtained lesion is characterized by edema, erythema, and accumulation of polymorphonuclear cells. When the antigen is injected, a complex is formed with the antibody, the stimulation of the complement occurs, and anaphylatoxins are generated rapidly, which causes granulation of the mast cells.

The intravascular local complex can also cause platelet aggregation and release of vasoactive amines that lead to an increase in swelling, erythema, and the formation of chemotactic factors, which causes the influx of polymorphonuclear leukocytes (PMNs) [28].

### 2.3. Induction of Rat Paw Edema

Six groups of Wistar rats were used; tests were performed by administering PT at doses of 10, 20, and 40 mg/kg of body weight. The two positive controls used were IBU and PRED at doses of 200 and 10 mg/kg, respectively. IBU is a non-steroidal drug with antipyretic and analgesic properties that inhibits the synthesis of prostaglandins at a central and peripheral level, and cyclooxygenase 1 and 2 enzyme isoforms (COX-1 and COX-2). PRED is one of the most used corticosteroids in the medical clinic that prevents or inhibits inflammation and immune responses when administered in therapeutic doses. A 0.1 mL of an antiserum solution of rabbit ovalbumin diluted 1:3 in 0.9% NaCl was injected into the sole of the right leg of the hindquarters of the rat. The contralateral leg was injected with 0.1 mL of a 0.9% NaCl solution, used as negative control, egg albumin was immediately injected intravenously at a dose of 25 mg/kg of body weight. In each animal, the volume of the leg edema induced by the antigen–antibody reaction was determined. The volumes of the swollen and control legs were measured 3 h after injection with a digital plethysmometer (LE-7500, Labequim S. A. de C. V.). Paw edema was determined by the difference between the value of the volume of the treated leg and that of the control [29].

### 2.4. Pleurisy Assay

For the Pleurisy assay, we used 25 Wistar rats organized in five groups of 5 rats each. The animals were injected in the pleural cavity with 0.2 mL anti-ovalbumin antibody diluted 1:10 in a 0.9% NaCl solution. Twenty minutes after the administration, a group was injected intravenously with 25 mg/kg of bovine albumin, the positive control group was administered PRED (10 mg/kg), the last three groups were treated orally with 10, 20, and 40 mg/kg of PT, respectively, and the negative control group with NaCl 0.9% solution. The animals were euthanized by CO_2_-inhalation 6 h after pleural inoculation; the volume of pleural exudate in the pleural cavity was quantified, and then centrifuged at 1500 rpm for 5 min, the sediment was placed on a slide, fixed with methanol, and stained with Giemsa for 10 min. A differential counting of neutrophils and lymphocytes was made [30].

### 2.5. Mouse Ear Edema Model

For this, we used 25 NIH mice organized in five groups of 5 mice each, and 2.5 µg of TPA dissolved in 20 µL of acetone was applied, according to the method of Young et al. [31], to both the internal and the external surface of the right ear of the mouse. After 1 h, PT was applied to the ear at different doses, 0.010, 0.020, and 0.040 mg/ear, each dissolved in 20 µL of acetone. We used indomethacin (IND) (0.5 mg/ear) as a positive control group. The mice were euthanized by cervical dislocation after 4 h, then a 7 mm diameter slice was made, and the central portions of the ears were weighed. The edema value was calculated by the weight difference between the treated ears (the ones on the right) and the non-treated ears (the ones on the left). The inhibition of edema (expressed in percentage) was also calculated versus the control group [31]. The results are presented as the average of the values obtained for each batch of animals ± standard error.

### 2.6. Myeloperoxidase Inhibition

For the determination of the activity of the myeloperoxidase enzyme (MPO), the inflamed ears were homogenized, according to Suzuki’s technique [32]. The absorbance was measured at 665 nm in a Perkin Elmer Lambda 3 spectrophotometer (Perkin Elmer Inc., Waltham, MA, USA). The enzymatic inhibition (expressed in percentages) corresponds to the absorbance differences observed with respect to the control group. The results are presented as the average of the values obtained for each batch of animals ± standard error.

### 2.7. Statistics

The statistical analysis of data obtained from the different anti-inflammatory assays was performed with an ANOVA followed by the Tukey’s multiple comparisons test, using Graph Prism 9.1.0 (GraphPad, San Diego, CA, USA)

## 3. Results

The three PT doses produced an inhibition of 51, 66, and 70%, respectively, of the rats’ leg edemas, giving a statistically significant difference and a 55% increase when PT-40 was compared to the negative control group (administered with NaCl 0.9%, 0.5 mL), and up to a 18% increase in inhibition compared to the positive control (IBU) (Figure 1).

In the assessment of the percentage of neutrophils, PT showed values of 60, 51, and 43% for the three doses, respectively, revealing a significant difference of up to 36% for PT-40 when compared to the negative control group, and very close to the values of the positive control group (PRED) (Figure 2).

The effect of PT on the content of lymphocytes in the pleural cavity exudate, induced by the antigen-antibody reaction, yielded values of 36, 42, and 39% for the three doses, respectively, with an increase of up to 28% for PT-20 when compared to the negative control group (NaCl 0.9%) and a 21% increase with respect to the positive control (PRED) (Figure 3).

The effect exerted by PT on the reaction volume for the antigen–antibody interaction induced in the pleural exudate showed values of 3.1, 2.7, and 3.3 mL, respectively, for each dose, observing a statistically significant decrease of up to 52% for PT-20 when compared to the negative control group (NaCl 0.9%), and very similar values to those yielded in the PRED (positive control) (Figure 4).

Evaluating the activity exerted by PT on the TPA-induced edema in the mouse ear, an inhibition of 72, 75, and 81% for each dose, respectively, was observed, and a 9% increase was observed when the PT-D3 group was compared to indomethacin (positive control) (Table 1).

Table 2 shows the results relative to the myeloperoxidase assay. A significant inhibitory effect is observed with the tested doses of PT; the high dose of PT (1.5 mg/ear) gives a slightly higher inhibition percentage than that observed with indomethacin.

## 4. Discussion

This study evaluated the anti-inflammatory and antioxidant capacity of pteropodin in rodents, which is a component of *U. tomentosa* that can modulate the immune system and inflammatory processes and it is used in traditional medicine to treat arthritis, among other diseases. Our results showed that PT (99% pure) exerts a potent anti-inflammatory effect in an acute inflammation model in rodents.

At the level of the collective consciousness of society, there is a rooted belief that everything “natural” is good, regardless of the amount consumed, since, if it comes from nature, it is considered that it will not cause any harm. In addition, the population usually does not associate medicinal and phytotherapeutic plants with the concepts of drugs and medicine, understanding drugs to be the substances that cause an effect in the organism depending on the dose, route of administration and interindividual variability. Regarding medicinal plants, it is often unknown how and where it was collected, its composition and uniformity, and the “dose” administered. Both medicinal plants and phytomedicines have been used (and abused) for their pharmacological properties and pleiotropic effects (broad, poorly selective) and many times without identifying the possible adverse effects and drug interactions produced by them [33].

In terms of quality, there are several difficulties: heterogeneous presentations in their composition, with multiple phytochemical components, many of which do not present well-characterized biological activities and more than one may contribute to the effect. To this, we must add the variation in purity and composition batch-to-batch given the natural variability of the plant of origin and the preparation methods, as well as the little knowledge of the stability of these preparations. Therefore, it is necessary to guarantee the quantifiable and uniform content of active substances in phytomedicines through processes that are harmonized. This is a basal condition to later be able to ensure the desired pharmacological effect, knowledge of its pharmacokinetic behavior, establish doses and therapeutic regimens, and reduce adverse effects [34].

Phytopharmaceuticals are medicines whose active substance contains the extract of a certain plant, unlike a chemical drug that comes from a chemically synthesized molecule. In the UC plant, there are several fractions of phytopharmaceuticals, among which are the pentacyclic oxindolic alkaloids and tetracyclic oxindolic alkaloids. For example, the pentacyclic oxindolic alkaloids of *Uncaria tomentosa* such as pteropodin induce the release of the factor regulator of lymphocyte proliferation in human endothelial cells, a property not attributable to tetracyclic oxindole alkaloids, quite the contrary, since they seem to reduce the activity of pentacyclic alkaloids in a dose-dependent manner in these cells. The tetracyclic oxindole alkaloids act on the central nervous system, while the pentacyclic ones act on the immune system, and both groups of compounds are found in two different chemo-types of the plant. Since the mechanism of action of tetracyclic and pentacyclic oxindolic alkaloids can be antagonistic to each other, it is of great importance to determine the chemotype through the analysis and adequate standardization of the plant to establish a specific effect of the active principle isolated from this fraction, which would give us the guideline to investigate its mechanism of action as a pure compound. On the other hand, using the pentacyclic oxindolic extract of the plant implies that the purity is lower since it includes pteropodin plus other compounds such as mitrafylline, isomitrafylline, isopteropodin, and uncarins. This mixture of compounds would give the therapeutic effect, but it would be difficult to know which of the components exerts the beneficial effect, hence the importance of studying pure pteropodin [35].

Recent research shows that the presence of tetracyclic oxindole alkaloids (TOA) inhibits the immunomodulatory effect of pentacyclic oxindole alkaloids. In recent years, a chemotype of *U. tomentosa* (Willd) DC has been found that does not present tetracyclic oxindole alkaloids (TOAF chemotype, “TOA-free chemotype”). The first clinical evidence indicates that this new TOAF chemotype could have great therapeutic potential as an immunomodulatory plant, which should be confirmed by the scientific community in the coming years [20,22].

The anti-inflammatory activity of cat’s claw (*U. tomentosa*) has been attributed, at least in part, to the inhibitory activity on cyclooxygenase-1 and -2 [36]. This anti-inflammatory action has been related to the capacity of the cat’s claw to neutralize the harmful effect of oxidizing organic substances, as well as its capacity to inhibit the expression of certain inducible genes during the inflammatory process [37].

The cortex of *U. tomentosa* (Willd) DC also has immunostimulant properties. The pentacyclic oxindole alkaloids increase the phagocytosis of macrophages and granulocytes and stimulate the proliferation of lymphocytes. In addition, cat’s claw causes macrophages to produce interleukins-1 and -6, which initiate the cascade of defensive activities of the immune system [38].

Several extracts of *U. tomentosa* root cortex have been tested for anti-inflammatory activity in a carrageenan-induced rat paw edema, and the quinovic acid-3-β-O-(β-D-quinovopyranosyl)-(27,1)-β-D-glucopyranosyl ester was isolated as one of the active compounds, which reduces the inflammatory response by 33% at 20 mg/kg. There is evidence that the combination of compounds is responsible for the strong anti-inflammatory effect of the extracts [39].

The addition of 100 μg/mL of an undefined extract of the stem cortex significantly attenuates the peroxy-nitrite-induced apoptosis in HT29 (epithelial cells) and RAW 264.7 cells (macrophages) (*p* < 0.05) and inhibits the expression of the lipopolysaccharide–induced nitric oxide synthase gene (iNOS), nitrite formation, cell death, and the activation of the nuclear transcription factor-қβ in RAW 264.7 cells. Oral administration of 5 mg/mL of the extract attenuates indomethacin-induced enteritis in rodents, reducing myeloperoxidase activity, morphometric damage, and liver metallothionein expression [40].

Anti-inflammatory activity of two types of extracts from the stem cortex: a hydroalcoholic extract containing 5.6% alkaloids (mainly of the pentacyclic type, extract A) and an aqueous freeze-dried extract containing 0.26% alkaloids (extract B) were assessed in the carrageenan-induced rat edema test. Extract A was significantly more active than extract B, suggesting that the effect could be due to the presence of pentacyclic oxindole alkaloids. Both extracts showed scarce inhibitory activity on cyclooxygenase-1 and -2. Only a slight inhibitory activity on DNA-binding of NF-қβ was observed [33].

The effects of a decoction of the stem cortex (10.0 μg/mL, lyophilized) on the production of tumor necrosis factor-α (TNF-α) and cytotoxicity in murine macrophages stimulated with lipopolysaccharide (RAW 264.7 cells) was assessed in vitro. The decoction prevented oxidative and ultraviolet irradiation-induced cytotoxicity and, at 1.2 to 28.0 ng/mL, suppressed 65–85% TNF-α production (*p* < 0.01) [38]. Cinchonain Ib, a procyanidin from the stem cortex, has an anti-inflammatory effect because it inhibits the activity of 5-lipoxygenase (≥100%) at 42.5 μmol/mL [38].

The anti-inflammatory activity of PT was evaluated with a widely used method for assessing anti-inflammatory substances, such as TPA-induced ear edema. The inflammatory process triggered by the topical application of TPA is due to the activation of the protein kinase C of the skin (PKC), which starts the inflammatory response. All anti-inflammatory agents show activity in this model, but mainly the dual COX/LOX inhibitors [40]; furthermore, cyclooxygenase (COX) inhibitors seem to be more effective as lipoxygenase inhibitors (LOX) than others in reducing the edematous response [41]. Along these lines, PT could behave like the COX inhibitors in this model.

PT produced a growing anti-inflammatory activity according to the administered doses of each fraction. It also showed important anti-inflammatory properties by significantly inhibiting the acute-induced edema dependent on the TPA dose, with an inhibition comparable to that presented by indomethacin with the same dose (500 µg/ear).

The present study is one of the few studies on PT and of the first regarding the assessment of its anti-inflammatory activity. These results, along with those obtained by Okada et al. [10] with *U. tomentosa* extracts, constitute the first reports showing extracts of a phytopharmacological genus with anti-inflammatory activity in vivo.

## 5. Conclusions

The results of this study support the use of PT in traditional medicine for the treatment of inflammatory diseases like rheumatism. In addition, it considers PT as one of the main compounds in plant extracts responsible for anti-inflammatory and antioxidant activities, which is easily obtained with excellent yields, making it very promising for the effective development of herbal extracts with pharmacological effects. Results are promising and encourage further studies on this active compound to assess it in other models of inflammation, both acute and chronic, and to determine the possible mechanisms involved in its pharmacological effects through its evaluation against specific mediators of inflammation, such as prostaglandins, nitric oxide, myeloperoxidase, and tumor necrosis factor, in addition to examining its ability to act as a scavenger of free radicals.

## Figures and Tables

**Figure 1 metabolites-13-00907-f001:**
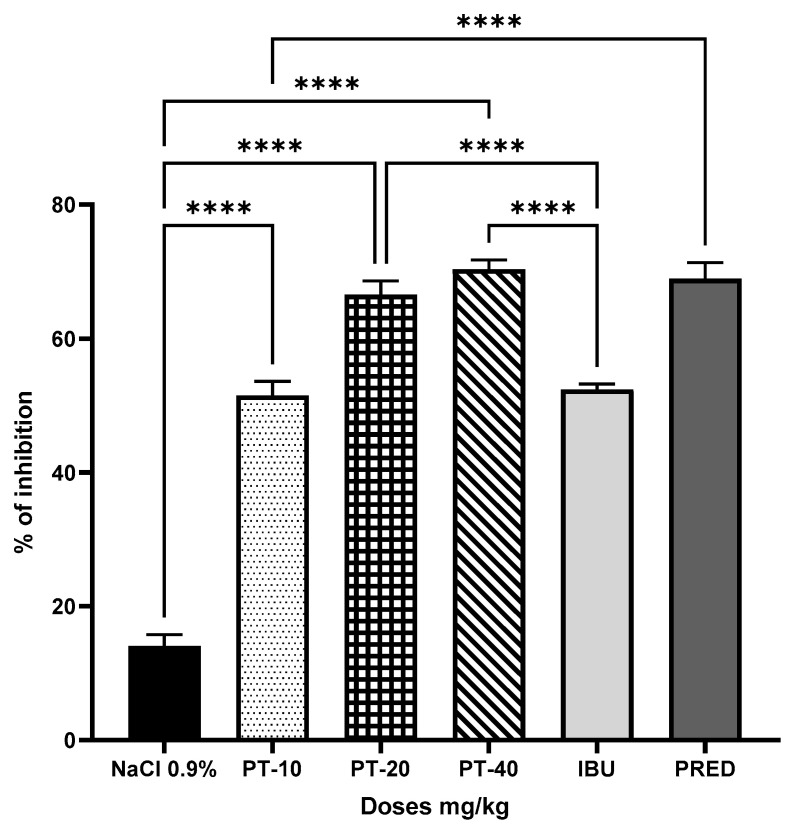
Anti-inflammatory effect of pteropodine (PT) in the rat paw edema model. The results represent the mean ± SD (*n* = 5), **** *p* < 0.0001, based on ANOVA followed by Tukey’s multiple comparisons test.

**Figure 2 metabolites-13-00907-f002:**
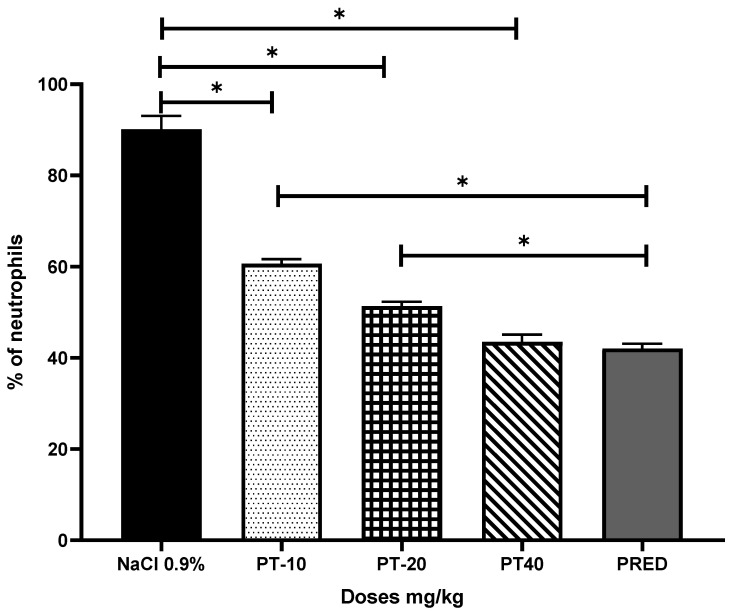
Effect of pteropodine (PT) on the neutrophils content of the exudate of the pleural cavity induced by the antibody–antigen interaction. The results represent the mean ± SD (*n* = 5), * *p* < 0.0001, based on ANOVA followed by Tukey’s multiple comparisons test.

**Figure 3 metabolites-13-00907-f003:**
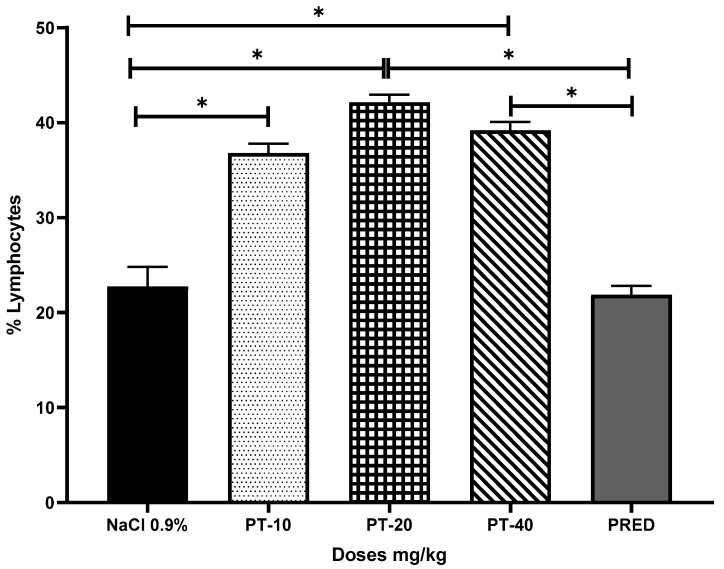
Effect of pteropodine (PT) on the lymphocyte content of the exudate of the pleural cavity induced by the antibody–antigen interaction. The results represent the mean ± SD (*n* = 5), * *p* < 0.0001, based on ANOVA followed by Tukey’s multiple comparisons test.

**Figure 4 metabolites-13-00907-f004:**
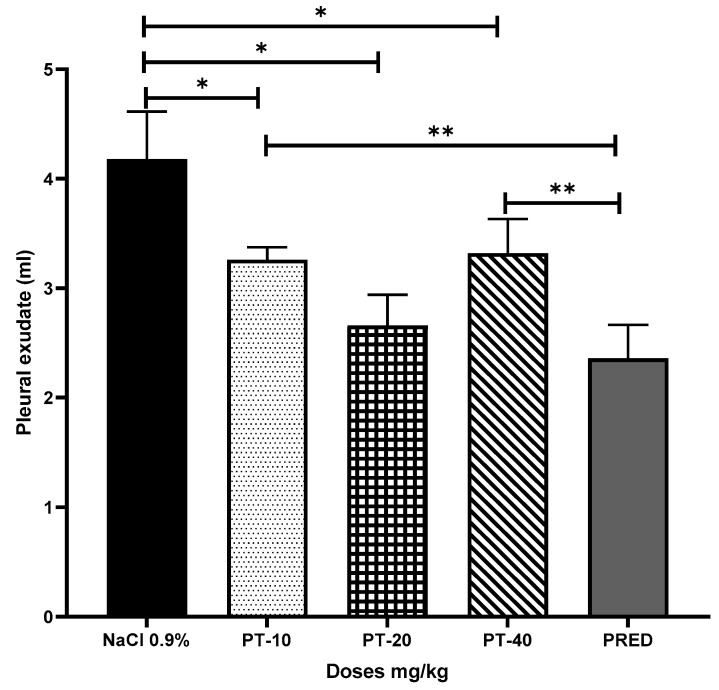
Effect of pteropodine (PT) on the volume of the pleural exudate induced by the antibody-antigen interaction reaction. The results represent the mean ± SD (*n* = 5), * *p* < 0.01, ** *p* < 0.001, based on ANOVA followed by Tukey’s multiple comparisons test.

**Table 1 metabolites-13-00907-t001:** Inhibition of the TPA-induced ear edema.

Group	Dose (mg/ear)	Edema (mg)	Inhibition %
Control TPA	0.0025	23.7 ± 0.3	0
Indomethacin	0.5	6.5 ± 0.12	72.6 *
PT-D1	0.010	5.7 ± 0.69	75.3 *
PT-D2	0.020	4.4 ± 1.5	74.2 *
PT-D3	0.040	3.1 ± 0.93	81.4 *

The results represent the mean ± SD (*n* = 10), *p* < 0.001. * Statistically significant difference with respect to the value obtained in 12-O-tetradecanoylphorbol-13-acetate (TPA) treated animals, ANOVA and Student’s *t*-test (*p* ≤ 0.05).

**Table 2 metabolites-13-00907-t002:** Inhibition of myeloperoxidase in the TPA-induced edema test.

Group	Dose (mg/ear)	Inhibition %
Control TPA	0.0025	0
IND	0.50	94.66 *
PT-D1	0.5	47.59 *
PT-D2	1.0	72.24 *
PT-D3	1.5	97.19 *

The inhibition percentage was calculated according to the equation %I = DAc-DAp × 100/DAc, where DAc is the arithmetic mean of the control absorbance, and DAp is the arithmetic mean of the tested agent absorbance. The obtained results represent the mean ± SD of 5 mice per group. * Statistically significant difference with respect to the value obtained in 12-O-tetradecanoylphorbol-13-acetate (TPA) treated animals; ANOVA and Student t-tests, *p* ≤ 0.05. IND = indomethacin.

## Data Availability

Data are contained within the article.

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
