# Peer review of "Anti-Inflammatory Potential of Pteropodine in Rodents"

_metabolites, 2023, doi:10.3390/metabo13080907_

Round 1

Reviewer 1 Report

This is an interesting study and the authors have completed most of the experiments from the animal studies. The experimental results are generally well written and structured. However, there are some shortcomings in regards to the legend presented in the figure, and I suggest that the authors should give supplements and corrections for improving the manuscript.

1.     In Figure 1, the control positive
It is not clearly specified item, the control group (ibu ?) might be better for the presentation.

2.     Figure 2, the control value, and Figures 3 and 4, the control value and control positive
The question is similar to the above.

3.     The manuscript does not provide any information about the antioxidant study from the scavenging free radical activity in blood.

4.     Also, the inflammatory factors in blood, such as cytokines IL-1beta of IL-6.    

Minor editing of English language in the figure legend is required.

Reviewer 2 Report

The article entitled " Anti-inflammatory and Antioxidant Potential of Pteropodine in Rodents " is supposed to treat the anti-inflammatory and antioxidant effects of Pteropodine on rodents. This study is very interesting in view of the very sensitive and important subject of pain in diseases such as cancer and rheumatoid arthritis.

I present here my remarks and comments:

1.          Introduction.

The introduction is well written and go around the issue of inflammation and pain due to rheumatism with a good bibliography.

2.         Material and method:

The biological tests chosen to show the anti-inflammatory effect of Pteropodine (PT) are well chosen and convincing to confirm the anti-inflammatory effects. But the authors should have sought the advice of a pharmacologist accustomed to applied experimental pharmacology. In this type of study, the concentration-response curve is used to determine the pharmacological parameters that allow to compare biological effects of the active principles used in the study. We would have liked to see the maximum and minimum concentrations that induce biological effects with IC50 and if possible the duration of anti-inflammatory effects. This information is very important for comparing bioactive natural molecules with drugs used in clinics such as Ibuprofen and prednisone.

3.         Results

Review the presentation of results using dedicated software such as GraphPad.

            Discussion

Very good discussion, but the results do not allow to do better by giving practical information that could lead to consider Pteropodine as a future candidate drug against pain and inflammation.

Reviewer 3 Report

The article talks about the anti-inflammatory and antioxidant effects of Pteropodine in reducing edema. The study is interesting and the authors shared some good data. The surprising result they found is that PT worked almost as good as a medically approved drug, in fact, better in some cases. Do you think that one compound would be that effective on its own if it is to be used alone? How would you relate the feasibility of using such plant compounds as therapeutic against edema?

Reviewer 4 Report

1. The title should be revised.

2. Abstract:

(1) “This study addresses the anti-inflammatory and antioxidant capacity of pteropodin in a murine model of arthritis and induced edema of the mouse ear.” It was not described exactly.

(2) Please check the data of the results “...52 to 74%...”.

(3) The results of the lymphocyte content is not included.

3. Introduction: Line 91-94: Please note rat and mice were both used in this study.

4. Materials and Methods: Please check the description of grouping, including the number of groups, the positive drugs used, and the dose of PT.

5. Results: Line 168-170: “...a 55% increase when compared to the placebo group, and up to a 18% increase in inhibition, compared to the positive control...”, which group compared to placebo group or positive control? Similarly, there are confusing descriptions in other paragraphs of the results part. Additionally, please check the results data “...3.1, 2.7, and 3.3 mL...” (Line 192).

6. Discussion: The authors should discuss and interpret the results based on previous related studies. Please check the correlation between the cited literature in Line 240-283 and your results, and place them in appropriate places in order to explore the significance of the results of your work.

Importantly, the authors did not study the action mechanism of PT as no any molecular indices were detected. Overall, I don't think this manuscript is appropriate for publishing in this journal.

It can be improved.

Round 2

Reviewer 2 Report

For information. I did read the authors' response to the thread. It is obvious that the process of developing chemical drugs can take 10-12 years and cost millions of dollars. I don't think the purpose of this work is to make a chemical medicine because no lab will put money for natural molecules that can't be patented. What I am talking about is following the WHO referential for the development of phytomedicines from traditional pharmacopoeia. To do this, it does not take millions of dollars or decades to get there. 

Author Response

We sincerely appreciate the comment and fully agree that our intention is only to contribute with scientific evidence to support the possible use of this compound as an alternative to the existing treatments that have been validated by the FDA. We are aware that it is necessary to create and apply a formal framework of use for phytomedicines, comparable to that of synthetic medicines, which implies very large and long-term studies under the recommendations of the WHO. It is also necessary to train prescribing professionals and the role of clinical pharmacologists in the generation and analysis of quality, efficacy, and safety evidence of these new phytomedicines. All these actions, together with the application of a regulatory framework for registration and the harmonization of quality standards for phytomedicines, will synchronously contribute to demolishing the belief that everything "natural is good" and to consider aspects related to its indications, dosage, security, and interactions.

Reviewer 4 Report

The manuscript has been improved, with just a few minor points outlined below.

1. Please revise the title to “Anti-inflammatory Potential of Pteropodine in Rodents”, As you said, the antioxidant effect of PT was evaluated by determining the myeloperoxidase enzyme activity, but this index was only measured in mouse ear edema test. Additionally, myeloperoxidase enzyme is secreted by neutrophilic granulocytes, and its activity reflect the degree of neutrophil infiltration, therefore, it is also an inflammation-related index. Please revise the abstract and main text correspondingly.

2. As you stated in Line 202 “The three PT doses produced an inhibition of 51, 66, and 70%, respectively, of the rat’s leg edema”, why in Abstract ...52 to 74%...? Additionally, in abstract, “52% of reduction of the pleural exudate volume, and low neutrophil count (28% or 36%?), the content of lymphocytes increases of up to 28%, when compared to the control group. We also found a 81% of inflammatory inhibition of the edema ear... this descriptions were confusing because you did not specify which group compared to the control group. Therefore, please revise the abstract carefully.

3. Line 122-123: ...determine its anti-inflammatory potential in a rat model that has the physiopathological mechanisms of rheumatoid arthritis and in a mouse ear edema model.

4. Line 166: “For the Pleurisy assay, we used 25 Wistar rats organized in five groups of 5 rats each.”

5. Line 239-241: “Evaluating the activity exerted by PT on the TPA-induced edema in the mouse ear, an inhibition of 75, 75, and 81 %, respectively, for each dose, was observed, and a 9% increase was obtained when compared the PT-D3 group to indomethacin (positive control)  (Table 1).”

Minor editing of English language required.

Author Response

Thank you for the observations and indeed, the activity of the myeloperoxidase enzyme is used as a marker of inflammatory processes in experimental animal models. Likewise, oxidative stress is currently proposed as a potential inducer of inflammation, with implication in the development of chronic pathologies at a systemic level, for which the determination of this enzyme was proposed as an indirect indicator of oxidation, however we have analyzed your coments and we consider that observation is adequate to only consider the result as an evaluator of the inflammatory process. Therefore, we decided to change the title to "Anti-inflammatory Potential of Pteropodine in Rodents" and focus the results on the evaluation of the anti-inflammatory activity of PT.